# Lock-In Thermal Test Simulation, Influence, and Optimum Cycle Period for Infrared Thermal Testing in Non-Destructive Testing

**DOI:** 10.3390/s23010325

**Published:** 2022-12-28

**Authors:** António Ramos Silva, Mário Vaz, Sofia Leite, Joaquim Mendes

**Affiliations:** 1Faculty of Engineering, University of Porto, 4200-465 Porto, Portugal; 2INEGI—Institute of Science and Innovation in Mechanical and Industrial Engineering, 4200-465 Porto, Portugal; 3CINTESIS—Center for Health Technology and Services Research, 4200-465 Porto, Portugal

**Keywords:** thermography, nondestructive testing, lock-in, simulation, laboratory tests

## Abstract

Lock-in thermal tests (LTTs) are one of the best ways to detect defects in composite materials. The parameter that most affects their performance is the cycle period of the stimulation wave. Its influence on the amplitude-phase results was determined by performing various numeric simulations and laboratory tests. The laboratory tests were used to infer part of the simulation parameters, namely the input and output heat, corresponding to the stimulation and natural convection. The simulations and the analysis of their results focus on the heat flow inside the sample and the manner they change for different geometries. This was performed for poly(methyl methacrylate (PMMA) and carbon fiber-reinforced polymers (CFRPs). The simulation of these materials was also used to create prediction surfaces and equations. These predict the amplitude and phase for a sample with a thickness *l* and a cycle period. These new findings were validated with new laboratory tests and two new samples. These validated the prediction surfaces and equations and can now be used as a reference for future works and industrial applications.

## 1. Introduction

### 1.1. Background

Non-Destructive Tests/Testing (NDT) is characterized by the ability to test components without diminishing their service period or degrading their performance. Field view techniques allow the examination of the area of a component, thus increasing the test speed. Most techniques use wave propagation like ultrasounds, visible, ultraviolet, X-rays, or infrared to test a component area. Infrared thermography measures the radiation naturally emitted by objects near the ambient temperature and creates temperature images using this information. External energy sources are used to amplify or create thermal patterns that provide insight into the component state external energy sources are used. These will provoke a temperature to unbalance. Studying and analyzing these patterns and how they evolve during the test enables the determination of some conditions and the identification of defects. Using optical stimulation (light) usually leads to good results and a setup that is easy to arrange with good portability, along with the good results provided. This type of stimulation is mainly used in two manners: (i) a simple uniform stimulation applied during a period (transient thermal tests) and (ii) a cyclic stimulation modulated as a sinusoidal wave. These last ones are called Lock-in Thermal Tests (LTTs) [1,2].

The temperature response is a combination of three factors: (i) heat received by radiation (optical stimulation), (ii) heat transfer by radiation and convection with the air, and (iii) conduction inside the component. The radiated heat from the sample is small and is usually discarded. With the increase in temperature, the heat loss by convection will also increase, and vice versa, meaning the heat will not travel from the surface to the interior of the sample but will travel from the inside to the outside of the sample [3]. Less energy (shorter stimulation) results in less thermal variations and less heat exchange by convection since the temperature difference is lower. Likewise, lower temperatures lead to less heat flow by conduction. With this type of test it is possible to analyze the temperature throughout its entire duration. However, determining the amplitude and phase response leads to precise and sensitive results.

The amplitude and phase images are determined by comparing (a) the stimulation and (b) individual pixel temperature. These are used to calculate the amplitude and phase response for each pixel. Two maps (images) of the object’s thermal response are then created. These are sometimes called ampligrams and phasegrams Since each of these images is obtained using several temperature measures, the results contain less noise than the temperature images. However, special care is necessary when choosing and using these sinusoidal stimulations. If they are not perfect sinusoidal waves, significant errors can occur [4,5]. A higher number of measures (thermal images) is preferable when calculating the phase delay and amplitude response. The best situation or characteristic response has been found for stimulations with periods of a few seconds and some cycles. Longer cycles result in the application of more energy to the object under analysis. However, the longer stimulation disadvantage is the inherent duration of the global thermal test. In industrial applications, a longer stimulation means that the NDT and the entire maintenance or preventive operation will have a higher cost or might become a bottleneck.

Numeric simulations predict the behavior of structures or components under predetermined conditions [6]. These are particularly useful in situations where exact solutions are unknown or complex. Another alternative to study processes is to build prototypes and test them. Apart from the cost, this process can also be time-consuming. Simulations can show the inside of a component or structures difficult or impossible in experimental testing processes. Currently, numeric simulations are almost entirely run on computers, being implemented in several programming languages, such as Python, FORTRAN, C, and Matlab^®^, among others. Some companies commercialize specific programs to perform simulations; the most common are COMSOL^®^, SIMULIA^®^, ANSYS^®^, Nastran^®^, etc. Using a virtual environment to manipulate the results also facilitates the management, documentation, communication, and pre/post-processing of data.

The most common numeric simulation uses one of three methodologies: Finite Differences Method (FDM), Finite Elements Method (FEM), or Finite Volume Method (FVM) [7]. They are also called the classic choices to solve Partial Differential Equations (PDEs). FDM uses a local Taylor expansion to calculate the PDE. However, it has some difficulties for complex geometries [8]. FVM equations are integrated for each volume with linear variations inside, resulting in errors when using tetrahedral meshes or elements with unbalanced dimensions [9]. Usually, FEM uses the weighted Galerkin method, being well suited for complex geometries. The development of the FEM in the last years made them the most common choice in commercial simulation software [10]. Due to the robust formulation of the FEM and its flexibility for a given mesh, this was the selected method in this work. The analysis of the evolution and distribution of the temperature in the samples should use Fourier’s first law of heat conduction. Here, a heat flux (q) that enters a surface has a thermal conductivity (K) of the material that travels for a certain distance (x) transversal to the surface with a temperature (T) [11]. The meshing process is also critical, with a constant spacing in the nodal points distribution leading to an equilateral and balanced element, preventing significant errors and singularities in the shape functions.

The selection of the materials is an essential part of the design of any component, even more in the case of high-performance elements. One of the most used materials is composite materials, particularly carbon fiber-reinforced polymers (CFRPs). These have a very high stress-to-weight ratio, among other characteristics. However, they present some disadvantages such as being anisotropic, having a costly and complex manufacturing process, and their failure is unpredictable, among other things [12]. For the latter, the importance of NDT in CFRP is even higher, especially in the amount of energy applied.

### 1.2. Recent and Relevant Works

Recently, artificial intelligence has gained much attention from the research community. It is also being used to improve NDT methods. Al-Athel et al. present a hybrid thermography, computational, and Artificial Neural Networks (ANN) approach to characterize sub-surface defects in composite materials. Their *“defects consisted of grill holes and the simulation data is fed to the ANN”.* Their results are not on par with other applications of ANN but are promising [13].

Stoynova et al. performed numerical simulations of a thermal 3D model with artificial defects under a lock-in test. This is one of several works that used LTT to perform subsurface defect detection and characterization [14]. Other similar works were performed by Shrestha [15], Chulkov [16], Pitarresi [17], An, Y.K, [18], Zoecke [19], and Peng [20]. The usage of lasers as an excitation source is not the most common. The delamination problem was researched by Swiderski using a numerical and experimental approach. With a laser stimulation with a wavelength of 808 nm, a maximum power of 32 W, and a FLIR 7600 SC camera, they showed it is possible to detect delaminations on CFRP [21].

Lock-in thermography is mainly used to analyze carbon-fiber reinforced polymers (CFRPs) or glass fiber-reinforced polymers (GFRPs). However, at a macroscale, electronic circuits, especially Printed Circuit Boards (PCBs), can be considered composite materials. Here, the most desired feature is not the improvement of the mechanical properties but rather the electrical connection between two or more points. If these connections are damaged, the PCB is useless. The usage of (a) flash, (b) transient, or (c) lock-in thermography has been used by Stoynova to study the shape and better understand their root cause [22,23]. Other similar and relevant works, even if not using simulations but focusing on laboratory tests, were presented by Breitenstein [24], Brand [25], Andersson [26], Leppänen [27], and Hovhannisyan [28].

Flash, Transient, and lock-in thermography are the classic Infrared Thermography NDT techniques. These require an intense heat source (transient test) or a well-controlled and linear power source [2,5]. To overcome these drawbacks, frequency-modulated stimulations have been studied by Mulaveesala and by Rani [29,30]. The highly damped behavior of the heat waves in LTT leads to dispersion in the surface thermal patterns. Hedayatrasa studied the viability and sensitivity of using thermal wave radar (TWR) to inspect CFRP [31]. Here, analog frequency-modulated (sweep), discrete phase-modulated (Barker binary coded) waveforms, and frequency-phase-modulated (FPM) were considered. The 3D Finite Element Analysis (FEA) showed *“outstanding performance of TWR at relatively high excitation frequencies is highlighted, particularly when approaching the so-called blind frequency of a defect”*.

Despite most research centering itself on CFRP or GFRP, Philipp et al. studied the application of thermal diffusivity of semitransparent polymer films. They presented a theoretical heat loss model by convection and radiation in semitransparent film surfaces. They claim that *“the slopes method is valid for any semitransparent film in the thermally thin regime”* [32].

Despite the several works studying LTT with experimental techniques and numerical simulations, several issues are persisting/missing in the literature and are addressed in this work. Most works are difficult to compare due to their differences in the test settings or do not present this information at all. This work aims to analyze the influence of the cycle period in LTT applied to CFRP samples for different thicknesses. An in-depth analysis of the sample’s heat flow and temperature variations is presented. The analysis of the temperature profiles at the sample surface for different cycle periods and the temperature profiles’ inversion pattern (related to blind frequencies) is exceptionally detailed. To summarize and facilitate the inclusion of the presented findings, prediction equations are presented. These predict the amplitude and phase results according to the component thickness and cycle period used. For this purpose, a numerical model was created using FEM and validated with poly(methyl methacrylate) (PMMA) and CFRP samples. This document and the work followed the structure presented in Figure 1.

## 2. Materials and Methods (Simulating LTT)

The present chapter presents the analyses and simplification of a 3D sample into a 2D surface. A rapid description of the equations implemented for the FEM and the mesh evaluation is shown. A comparison between the temperature results from the laboratory tests and simulations is also present.

CFRP is anisotropic material built from overlapping sheets of carbon with resin as a bonding element. The initial LTT was done with a sample made from PMMA and used as a reference and had the geometry presented in Figure 2. In this 3D representation of the designed sample, the thickness and slot width were increased to improve visualization. The overall measures are 210 × 160 × 4.5 mm, with slots having 10 mm in width and 0.5, 1.5, 2.5, and 3.5 depth. A mate varnish (kameralack) from Tetenal^®^ is used to paint the transparent PMMA. This provides a high and uniform emissivity (0.98), thus enabling the usage of optical stimulations. This procedure was also used in previous works [33].

The last analyses refer to a CFRP sample. Its size is 206 × 170 × 8 mm, built using 28 carbon fiber sheets, and has circular blind holes. There were twelve blind holes with a geometry that combined diameters of 10, 16, 20, and 25 mm, along with depths of 2, 4, and 6 mm.

The laboratory tests consisted in applying an optical stimulation modulated as a sinusoidal wave. The thermal camera is positioned horizontally and perpendicular to the sample at approximately 1 m. The halogen lamp was 0.8 m above the thermal camera. The stimulation and thermal measures were executed on the sample side without visible slots, using the configuration known as reflection mode [34]. In the initial test, a stimulation period of 20 s during 15 cycles was applied. The stimulations were applied using a halogen lamp from Hedler^®^, model H25S, with 2500 W. These were controlled using the power box model from AT—Automation Technology GmbH^®^. The sample temperature was measured using a thermal camera from FLIR^®^ model 7500, with a measuring waveband ranging from 1.5 to 5.1 and less than 25 mK of Noise Equivalent Differential Temperature (NEDT). A computer synchronized the stimulation and the image acquisition through the application IRNDT V.1.7, also from AT—Automation Technology GmbH. These initial tests were used to optimize the simulation script and are mentioned further ahead.

It was intended to simulate the temperature variations in the sample with the slots having a uniform geometry, vertically oriented, for an isotropic material. Here, it is possible to discard the vertical dimension (Z dimension in Figure 2). This simplifies the geometry into a two-dimensional shape. This approximation was validated by analyzing the heat transfer during a sample LTT.

Depending on the temperature gradients, this simplification of the 3D sample into a 2D area, corresponding to the sample horizontal cross-section, might be incorrect. Thus, it was necessary to determine the Biot number. If this is less than 0.1, the simplification is valid. An accurate model to calculate the heat transfer coefficient was proposed by Churchill and Chu in 1975 [35]. This model requires the previous calculation of the Rayleigh number (Ra) and the Prandtl number (Pr) and can be calculated using Equation (Equation 4).

During the LTT, the sample is warmed, appearing as a vertical airflow due to the natural convection. By multiplying the Grashof number (Gr, Equation (Equation 1)) by the Prandtl number (Pr, Equation (Equation 2)), the Rayleigh number (Ra, Equation (Equation 3)) is obtained [36]. With temperature T∞ equal to the laboratory tests and Ts twice the maximum amplitude recorded in the initial tests, Ra equals 3.84 $\times 10^{6}$. It is a significant laminar flow (101 and 109), meaning the airflow due to convection is significant.

Equation (Equation 4), proposed by Churchill and Chu in 1975, determines the heat transfer coefficient [35]. With this coefficient h is calculated as a Biot of 0.08 (Equation (Equation 5)), thus being less than the established limit of 0.1. Indicating the vertical temperature gradients inside the sample due to the natural convection can be discarded, and any temperature patterns are a direct result of a defect (in this case, a machined slot). Therefore, simplifying the sample into a two-dimension horizontal surface is a good and correct approximation (Equation 1).
(1)$$GR=\frac{g\beta \left(T_{s}-T_{\infty }\right)L_{c}^{3}}{\upsilon^{2}}\to GR=5.39\times 10^{6}$$
(2)$$PR=\frac{C_{p}\mu }{K}\to PR=0.7$$
(3)$$Ra=GR\times Pr\to Ra=3.84\times 10^{6}$$
(4)$$h=\frac{K}{L_{c}}\left[0.68+\frac{0.67\times Ra^{1/4}}{{\left[1+{\left(0.492/Pr\right)}^{9/16}\right]}^{4/9}}\right]\to h=0.384\left(\frac{W}{m^{2}K}\right)$$
(5)$$Bi=\frac{hl}{K_{PMMA}}\to Bi=0.08$$
where:υ—Kinematic viscosity, $15.68\times 10^{-6}$$\left(\frac{m^{2}}{s}\right)$$T_{s}$—Temperature at the sample surface, 306.5 (K)$T_{\infty }$—Ambient temperature, 296.5 (K)β—Volumetric thermal expansion coefficient, 3.3 $\times \;10^{-6}$$\left(\frac{m^{2}}{s}\right)$$L_{c}$—Characteristic length, 0.16 (m)Cp—specific heat, 1005 $\left(\frac{J}{\mathrm{Kg}K}\right)$μ—Viscosity, 1.857 $\times \;10^{-5}$$\left(\frac{\mathrm{Kg}}{s\times m}\right)$*k*—Thermal conductivity, 0.0262 $\left(\frac{W}{m\times K}\right)$*l*—Sample thickness, 4.5 $\times \;10^{-3}$ (m)$K_{PMMA}$—PMMA Thermal conductivity, 0.21 $\left(\frac{W}{m\times K}\right)$

The simulation of a transient response is well described in the literature. A sequence of transient stimulations can be used to discretize a sinusoidal one, like the one used in lock-in tests. Fourier’s first law, in its basic form, must be expanded to match the tested sample geometries and the PDE. Since the Z axis is discarded, the geometry is defined in the X and Y coordinates and time (t). By expanding the PDE, commonly used in the finite element approximation (Equation (Equation 6)), we obtain the final and the system governing equation. The simulations were implemented in MATLAB^®^ and followed the methodologies described by Erik Thompson [11].
(6)$$\frac{\partial }{\partial x}\left[K_{x}\frac{\partial T}{\partial x}\right]+\frac{\partial }{\partial x}\left[K_{y}\frac{\partial T}{\partial y}\right]+Q_{i}=\rho \times Cp\frac{dT}{dt}$$
where:$Q_{i}$—interior heat source per unitary area (stimulation);$K_{x}$—conductivity coefficients in the **X** direction;$K_{y}$—conductivity coefficients in the **Y** direction;ρ—material relative density;Cp—Heat capacity.

The simulated area required the definition of boundary conditions that mimicked the laboratory tests. It applied a natural convection coefficient to the external contour, the blue line in Figure 3. Along with the natural convection (energy output), the stimulus or energy input was simulated, the green line in Figure 3. In these nodes, the heat generation coefficient (Qi), contrary to all the other nodes, was not zero. Due to the uncertainty in determining the heat applied by the halogen lamps, the remaining parameters were determined by fitting the temperature curves to the laboratory data (the results are in the next section).

For the simulated section, X has the range [0–210], Y has the range [0–4.5], and t ranges from zero up to 15 times the cycle period (in seconds). The sample horizontal dimensions a, b, and d were kept equal to the samples, more precisely with 40, 30, and 10 mm. The mesh had a uniform spacing in the Y direction (l and c). The surface was divided into quadrilateral shapes with one nodal point at each corner, thus four nodes per element. Some recommendations for the mesh size and shape are available in the literature and provide some guidelines [37,38]. Fine-tuning of the mesh was performed through several simulations with different meshes. The results are presented in Table 1. The mesh’s name indicates the number of elements in dimensions *d*, *a/b*, and *c* represented in Figure 3. The results in Table 1 show that there is a temperature conversion. In all the sets of meshes evaluated, the selected mesh is highlighted in bold, and the final mesh is in red.

The initial laboratory tests served to fine-tune the simulation parameters. Both correspond to a test with 15 cycles and 20 s of the cycle period. Figure 4 presents the temperatures inside the sample. These results are from a simulation at the end of the 15th cycle. Appendix A presents these same temperatures. In both, the Y scale was increased to improve visualization.

The temperature profiles from the laboratory tests and simulation (Figure 5) show a significant difference. In the laboratory tests, the maximum temperatures are higher than in simulations. This is due to the reflected radiation from the stimulation. For the lower stimulation amplitude, the reflected radiation is almost null. This makes the temperatures in the laboratory tests and simulations very close. As a result, the stimulation amplitude value was set to approximate the temperatures in the lower part of the cycles (valleys).

Observing the temperature profile at the last frame of the tests (Figure 6) is the second validation of the simulation parameters (Figure 6). From the laboratory test, slot one thermal profile base was slightly wider than the simulated one, and the maximum temperature for slot 4 was lower. Apart from these two differences, all the other temperatures were very similar. Therefore, these differences are considered irrelevant.

## 3. Simulation Results

The sample simulations had a sinusoidal stimulation, with each simulation having 15 cycles and 65 different cycle periods, ranging from 0.02 to 80 s. The analyses were performed for all of these; however, to facilitate the representation and its observation, only the results of 15 are presented. Figure 7 presents the temperature evolution of the central node from slot one, through the entire LTT. For shorter cycle periods, it is visible that the temperature evolution is a linear oscillating curve. Longer cycle periods present a temperature response that combines sinusoidal stimulation and a negative exponential evolution. The temperature curves for cycles of 20 to 80 s present higher temperature variations. These higher variations in the slot temperature occur in the first three to four cycles. After these, the slot temperature evolution is mainly due to sinusoidal stimulation. The sinusoidal temperature variation in the entire profile is uniform. Among the several simulations, the temperature variations in the center of slot 4 were considerably small.

Figure 8 represents the temperature profiles in the stimulus area after 15 cycles, also obtained with cyclic stimulations from 0.02 to 80 s per cycle. Slots 1, 2, 3, and 4 are visible for stimulations longer than 0.1, 0.2, 0.7, and 2 s, respectively. Increasing the stimulation period increases the overall temperatures. However, as the slots present higher temperatures, their profiles also become blurred, indicating an attenuation in the temperature distribution. For 15-s periods, the temperature response starts to invert the temperature response. The temperature difference between slot one and its boundaries starts to decrease. Increasing the cycle period makes the boundaries harder to identify. For stimulations longer than 32 s, the temperatures observed for the center of slots one and two are lower than those in its sound areas. Figure 9 presents the temperature profiles during the entire test with a cycle period of 40 s.

One of the main principles behind using the amplitude analyses is that any slight difference will be amplified (cumulative effect) at each cycle. Thus, observation of the temperature distribution after a certain number of cycles is a starting point to identify the defects in thermal patterns. The amplitude and phase delay calculus was executed at an application in LabVIEW^®^. It uses a Discrete Fourier Transform (DFT) to extract the relative gain and phase between the normalized sinusoidal reference and the temperature at each node. Figure 10a presents 15 of the 65 amplitude results to facilitate the visualization. The small temperature decays observed at the boundaries of the slot become visible for stimulations longer than 11 s (Figure 10a). A similar amplitude behavior is observed in slot 2. However, they are softer and tend to appear for slightly longer stimulations. The difference between slots 1 and 2 is the complete inversion of the amplitude profile at the slot 2 location, passing for a near “zero” identification for the stimulation of 14 s (Figure 10a). The thermal response of slot 4 is low, despite the cycle period used. We define the amplitude response as the temperature difference between a slot and its two sound areas. One can consider this to be technique sensitivity. Slot 1 amplitude response has a local maximum near the three-cycle period, a local minimum at the 11 cycle period, and starts to increase for stimulations longer than 15 s (Figure 10b). This behavior is observed for all the slots as visible in Figure 10b,c for different cycle periods. These local maximum and minimum result in a null response for the second and third slots. This can indicate the presence of blind frequencies. A stimulation frequency for which the slot presents zero thermal amplitude. This last aspect reveals the limitations of amplitude images. When comparing the amplitude response with the final temperature, the fourth slot was visible in the temperature profiles. This is a way to overcome the existence of blind frequencies.

Figure 11a presents the phase profiles for cycle periods from 0.02 to 80 s per cycle. With a cycle time of 0.02 s, the phase delay was considerably high and diminished for longer cycles. Slot 1 was once again the most visible, presenting the higher profile variation (increase in the phase delay) and appearing earlier (stimulation of 0.2 s). Slots 2, 3, and 4 were visible with stimulations of 0.4, 0.5, and 1 s, respectively. In Figure 11b and for slot 1, we observe a continuous increase in the phase difference. Since all the slot temperatures are a response to the stimulus, these present a time delay. Thus, all the phase differences presented negative values. Apart from this, the phase difference of slots 2, 3, and 4 present similar profiles to the amplitude difference. They have a local maximum followed by a local minimum and decrease with the increase of the cycle period. While not having this local maximum and minimum, slot 1 also presents a similar pattern.

Stimulations with a small amplitude response lead to phase images with high noise. These usually result from short stimulation periods and should be ignored. A slot can be observed as an area with a smaller thickness, thus, more heat per volume of material. However, the thicker area will have a higher damping factor, resulting in a slower dynamic. So, detecting and characterizing defects is accomplished by observing the different dynamics of the sound areas. The area corresponding to slot 1 is thinner than the sample thickness. Thus, slots 2 and 3 better represent the behavior of an “industrial” defect. Slot 2 and 3 behaviors are difficult to fully understand just by observing Figure 11. Thus, a detailed observation is presented in Figure 12. The evolution of the phase difference of the slots resembles the amplitude response. Slots 2 and 3 reached a local maximum with cycle periods of 4.5 and 6.5 s (Figure 12a,b). The phase delay decreases with the increase of the stimulus, being most visible in slot two and less visible in slot three. A local minimum is reached for slots two and three with cycle periods of 14 and 28 s, respectively. Slot 3 passes through an almost null response for a 36 s stimulation period. This is a blind frequency in the phase image. For slot two, the curve in Figure 12b does not cross the zero axis, being very close. With the existence of noise in laboratory tests, this slot might not be visible. However, slot 3 reaches a minimum of 0.01, leading to the intersection of the zero phase delay, thus the existence of two blind frequencies. These conditions that result in minimum phases should be avoided, and the thermal tests should be conducted with cycle periods that are shorter or longer. Longer stimulation will result in phase images with the contrast presented in Figure 11b.

## 4. Analyses of the LTT Simulations

The first aspect to highlight from the results is the difference between the laboratory and simulated temperature curves. The reflected radiation prevented a direct comparison between the two sets of data. Comparing the cycle’s lower temperatures (valleys) was the selected criterion to validate the FEA parameters. The comparison of the temperatures at the end of the test was used as a confirmation. Overall, the laboratory tests validated the settings used in the FEM simulations.

Using a thermal image during LTT is one manner to identify some defects. This analysis is relevant if the stimulation period is near a blind frequency. Using cyclic stimulation also has a cumulative effect and amplifies temperature profile patterns, showing different temperature patterns than the ones observed when using a constant and single stimulation. The samples used were made of PMMA. Its thermal conductivity is slightly lower than a CFRP but is fully isotropic. A material with low thermal conductivity is expected to increase its global temperature during cyclic stimulation. Thus, when a fast sinusoidal stimulation is applied, low-temperature variations are expected, and increasing the stimulation cycle period delivers more energy to the specimen, and higher temperatures will show. This type of behavior is observed in over-damped mechanical systems. Since all the tests used sinusoidal stimulations, it was impossible to separate the negative exponential from the sinusoidal response in the temperature data. When these present a constant sinusoid evolution, this indicates the test was long enough for the temperature to reach a steady-state regime. If longer sinusoidal stimulations are used, an exponential thermal response will appear, overlapping with the sinusoidal thermal response. Increasing the period of cyclic stimulation also increases the amount of energy received by the sample.

Thinner areas receive more energy per volume, resulting in higher temperatures and vice versa. For long cycle periods, areas have higher temperatures, and with the stimulation being in a lower instance of the cycle, these areas will start to warm their surroundings. This lateral temperature conduction (along the X-axis) will result in the dispersion of the temperature gradients, like in a transition area such as the slot boundaries. This is observed in the eight boundaries of the slots and at the profile extremities. As the cycle period increases, the time available for this lateral heat flux to become visible also increases. A higher difference between the slot temperature and the surrounding sound areas increases the lateral heat flux and consequentially softens the temperature profile. The resulting temperature profiles have blurred boundaries (difficult to identify), especially for long stimulations.

To optimize an LTT, a low number of cycles should always be used. Since most cameras can reach tens of frames per second, obtaining more than 100 images per cycle is an easy setup to accomplish. To calculate the amplitude image, it should use at least two cycles. Figure 13 presents the variations in the phase as a function of the number of cycles and cycle periods. For a short stimulation (cycle period of 5 s for example), the hardest slots to detect require more cycles, while the deepest slots reach the maximum phase difference with 5 to 9 cycles. With the stimulation of 30 s, this introduces blind frequencies and slots 2 and 3 detectability decreases with each cycle. This is particularly critical for slot 3, reaching a zero phase for nine cycles, while the third slot indicates that with a few more cycles this slot would not be detected. For longer stimulations (80 s for example), the smallest slots are mainly insensitive to the variation of the number of cycles. In contrast, the third and fourth slots present some big variations. The fourth slot presented a higher phase delay with stimulation of 15 s and one cycle when compared with the 15 cycles and stimulation of 80 s per cycle, which results in a considerably shorter test; however, this is not a usual situation and should not be used as a reference.

In LTT, it is impossible to identify a single image that produced the desired results. The best results (better detectability) occur when analyzing the amplitude and phase images. Thus, for these types of data/images, the optimal test parameters can be calculated. As such, we performed new simulations for PMMA and CFRP samples. These simulations focused on a sample with one slot (equivalent to slot 3) with various sample thicknesses and cycle periods. For each simulation, an amplitude and phase difference are calculated. Finally, the optimum cycle period was determined. For this, both phase and amplitude images were considered.

Figure 14 represents the amplitude variation as a function of the cycle periods (from 0.1 to 60 s) and for sample thickness ranging from 0.1 to 10 mm. Here, longer stimulations lead to higher temperature differences. When increasing the cycle period, a peak was not obtained, and the locations of the lower increasing rate were identified. These locations are represented by the green points in Figure 14. Due to the amplitude response stability (Equation (Equation 7)), this is considered the ideal location to conduct the LTT for PMMA samples. From the analyses, we concluded that the amplitude images do not provide the most accurate data. Despite a long cycle period leading to a higher amplitude result, the defect boundaries will become blurred. Therefore, the amplitude images provide a direct analysis and quality and accuracy control for the phase images. A higher amplitude response means that the temperature variations are significant and thus lead to less noise in the phase delay images. They can also act as a control preventing any false detection due to blind frequencies. Therefore, Equation (Equation 7) indicates the ideal stimulation for the amplitude analyses. However, it is less relevant than the ideal phase curve.
(7)$$CyclePeriod_{PMMAamplitude}\left(s\right)=4.1615+11.6358\times l-0.5648\times l^{2}$$

In previous chapters, the phase images were considered one of the most sensitive ones. Apart from the blind frequencies, these should always be used to assess the sample results. Figure 15 presents the phase differences between the reference slot and its sound areas. The “negative” areas reveal the crossing of the zero axis. So, at least one blind frequency should occur, corresponding to the inversion of the phase profiles. The surface in Figure 15 can be divided into three main areas: (i) the peak for thicker samples and short cycle period; (ii) the valley on the diagonal of the surface; and (iii) the maximum area for the thinner samples and longer stimulations. In area (i), the amplitudes are very low; thus, the phase images will be very noisy. The area that presents a valley (area (ii) is a direct result of the existence of a blind frequency. From the analyses of the several slots, we conclude that a cycle period producing a minimum phase delay of a defect could correspond to the exact blind frequency of another. As a result, cycle periods and sample thicknesses that lead to these locations should be avoided. Finally, area (iii) corresponds to the maximum obtained for long stimulus periods. This maximum area indicates the presence of a higher difference in the phase profiles. These points occur for long stimulations and samples thinner than 5 mm. The ideal stimulations should be here. A regression of these green points is presented in Equation (Equation 8). This is the recommended cycle period function of the sample thickness. The LTT ideal parameters depend on the method used to identify and characterize the defect. The ideal parameters should consider where the best phase or amplitude images are obtained. While the phase images tend to present higher detectability and signal-to-noise ratio (SNR), the amplitude response should be high enough to enable the phase image to eliminate most of the existing noise.
(8)$$CyclePeriod_{PMMAphase}\left(s\right)=2.029+8.694\times l-0.309\times l^{2}$$

The next step was to find the best parameters to conduct an LTT in CRFP. This is an anisotropy material, particularly for its thermal properties [39]. Different fiber orientations, layouts, and resign percentages also change its mechanical properties [40,41]. The simulations of the CFRP sample were the same as used previously, with thermal conductivities for the X and Y directions of 6.3 and 0.6, respectively [14]. It should be considered that the following results and analyses might be different for different properties such as different resin percentages or fiber alignment [42].

Figure 16 presents the amplitude of different functions of the sample thicknesses and cycle periods. The amplitude difference response presents a shape similar to the one observed in the PMMA simulations. Comparing both surfaces, the amplitude response for the PMMA sample is higher than for the CFRP. This may result from the different and higher thermal conductivity in the fiber direction and parallel to the surface, thus helping to blur and reduce the thermal patterns in the CFRP samples surface. Higher thermal conductivity decreases the temperature variations during the test. If only considering the amplitude images, using cycle periods indicated by the green dots is recommended. A fitting to these green points results in Equation (Equation 9).
(9)$$CyclePeriod_{CFRPamplitude}\left(s\right)=7.935+1.875\times l-0.0356\times l^{2}$$

One of the main tasks of this work is represented in Figure 17, the phase difference as a function of the cycle period and sample thickness for the phase response of CFRP samples. An obvious observation is the resemblance between the phase surface corresponding to the CFRP and the PMMA. Here, the surface also crosses the zero phase plane, revealing the existence of blind frequencies in some situations. Long cycle periods are also recommended, despite the higher values obtained with shorter cycle periods for thicker samples (surface valleys). In this manner, the low amplitude responses are avoided, and the noise in the phase images decreases. In the PMMA phase surface, when reaching the 5 mm thickness, the phase revealed the incapacity to maintain a good response. In the phase surface corresponding to the CFRP, the cycle period for which the maximum phase is almost proportional to the sample’s thickness. This is obvious in the best phase response given with Equation (Equation 10). Thus, it was obtained by a curve fitting the green points in Figure 17.
(10)$$CyclePeriod_{CFRPphase}\left(s\right)=2.15+5.53\times l-0.25\times l^{2}$$

## 5. Lock-In Thermal Test Validation

### 5.1. Validation of a Sample Made of PMMA

The previous chapter presented the best parameters for performing an LTT for PMMA and CFRP samples. The results show that certain conditions resulting in phase images can lead to inconclusive results. To prevent this, amplitude images should also be considered in the evaluation. The following validation tests used a different sample. It had four slots with a depth equal to slot three and widths of 5.0, 7.5, 10.0, and 12.5 mm. These validation tests consisted of laboratory tests LTT with various cycle periods.

Figure 18a,b illustrate the averaged amplitude and phase cross-section profiles obtained for the validation LTT. The amplitude response is proportional to the cycle period, as observed previously. Observing Figure 15, a small phase response was expected for shorter cycle periods as presented in Figure 18b. With the increase in the cycle period, the phase response in the slots’ evidence started to increase. Similar to the simulation pattern and the evolution described previously, the phase difference at the slots becomes constant. The downside of longer stimulations (60 s) is the fading/blur of the boundaries. In the phase profiles, this is evident. Apart from the possible identification of defects, these long cycle periods do not provide better results and should not be considered. For the sample of PMMA and using Equation (Equation 8), the ideal cycle period has 35 s. Observing Figure 18a, one can say that the ideal cycle period is between 30 and 40 s.

### 5.2. Validation of a Sample Made of CFRP

Performing an analogy to the observations of the PMMA sample, the overall thermal response for the amplitude images should increase with the cycle period. Figure 19a,b present the vertical profiles for two groups of slots, corresponding to the amplitude profiles of 2 and 4 mm thick (slots areas). According to the pattern described, the amplitude response in the slots should increase for long cycle periods. Naturally, a deeper slot and a long cycle period lead to better slot visibility. An important aspect is a similarity in the amplitude differences between 20 and 30 s. This indicates that the amplitude variation resulting from a possible increase in the cycle period is becoming smaller, as predicted by the observed amplitude surface for CFRP (Figure 16).

Figure 20 presents the phase profiles for the two sets of slots discussed. Similar to the PMMA samples and their phase, the CFRP profiles present a crescendo with the cycle period. The phase delay revealed low variations for the two shorter cycles, presenting a considerable variation for the 20 and 30 s profile. This pattern follows the behavior of the PMMA sample and the results of previous simulations. The phase delay for the cycle periods of 20 and 30 s are similar, thus validating the optimum cycle period of 26 s (Figure 20a,b).

## 6. Conclusions

This work intended to study the influence of the cycle period in LTT for CFRP samples for different thicknesses. This was accomplished using simulations and validated experimentally using PMMA and CFRP samples. The analysis of laboratory conditions determined if the assumptions and simplifications were correct. Here, the Biot number was significant and low enough for the intended simplification to be possible. The mesh optimization diminished the computational cost and allowed accurate results. Fitting the temperature results through time and at the end of a cyclic simulation with 15 sinusoidal cycles, it was possible to determine the stimulation parameters.

The temperature differences resulted from the stimulation reflected radiation and were impossible to remove from the thermal measurements. Apart from this expected difference between the laboratory tests and the simulation, the similarity was very high.

Numerous LTT simulations were performed, producing the amplitude and phase results for different sample thicknesses and cycle periods. Despite the current results, some different behaviors occur when changing the number of cycles. Even if not compromising the evaluations, the number of cycles might decrease the test sensibility. The initial stimulations provided a comprehension of the temperature through the entire thermal cyclic test, thus providing good visualization of the heat flow inside the sample, inaccessible experimentally. An important aspect is the signal-to-noise ratio presented by the amplitude and phase. The amplitude images and the resulting profiles revealed a lower signal-to-noise ratio than the phase images. From all of the data, the phase images demonstrated a higher sensitivity to the stimulus and cycle period. The results also show that a shorter stimulation reveals a better boundary definition. In some situations, the blind frequencies can result in a null response in the phase images, only visible in the amplitude response. The rapid variation in the phase profiles was also predictable.

The recommended cycle period from the prediction surfaces was validated experimentally with two samples. These are different from the ones used to create the mentioned surfaces. After various laboratory experiments, the temperature, amplitude, and phase results validated the previous equations and prediction surfaces relating to the PMMA and CFRP sample. It is worth mentioning the CFRP validation sample was 8 mm thick and had circle blind holes.

To present this work’s findings and facilitate their usage in future research, prediction surfaces and equations for the amplitude and phase difference were created. These consider PMMA and CFRP. The prediction surfaces can help in the identification of blind frequencies and prevent false negatives for a determined component thickness.

## Figures and Tables

**Figure 1 sensors-23-00325-f001:**
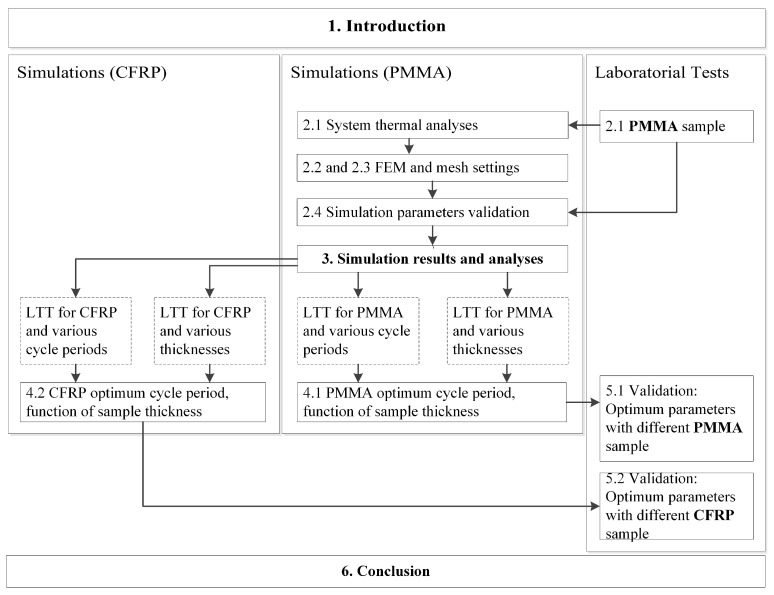
Workflow and article structure.

**Figure 2 sensors-23-00325-f002:**
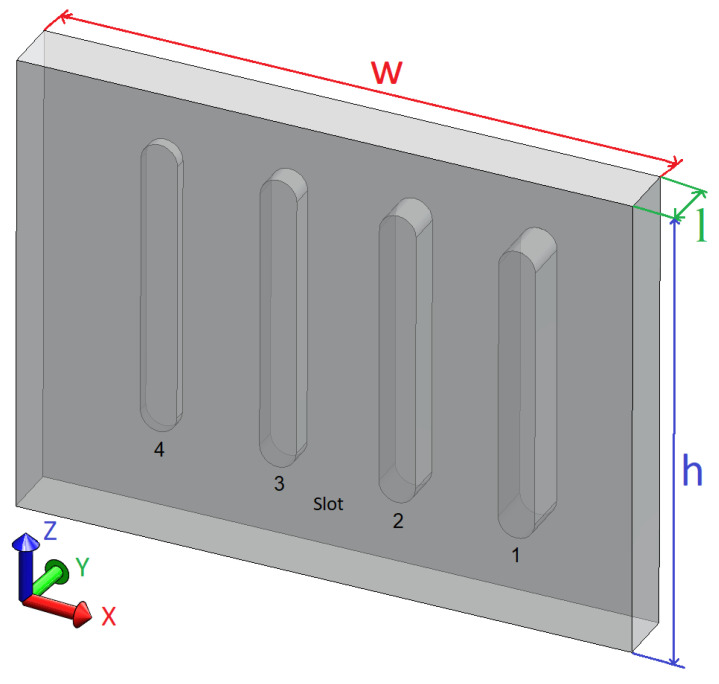
PMMA sample and its slots.

**Figure 3 sensors-23-00325-f003:**
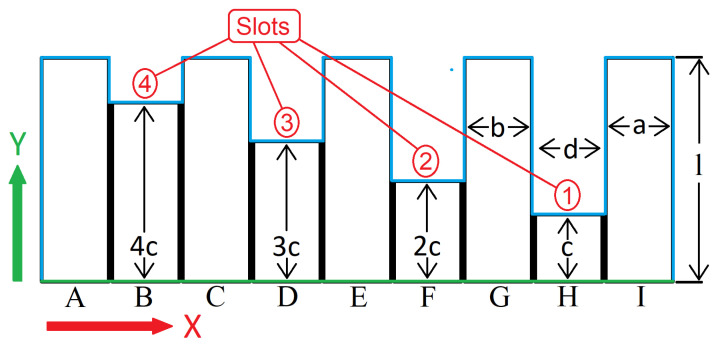
PMMA sample and its slots.

**Figure 4 sensors-23-00325-f004:**
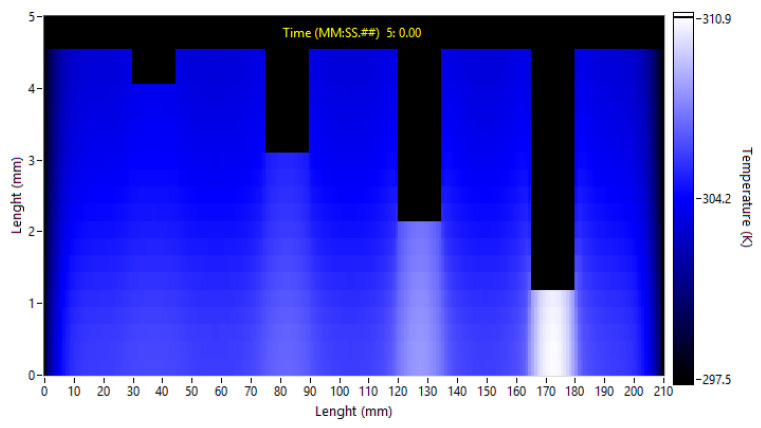
Temperature distribution after the initial simulation, 15 cycles of 20 s each.

**Figure 5 sensors-23-00325-f005:**
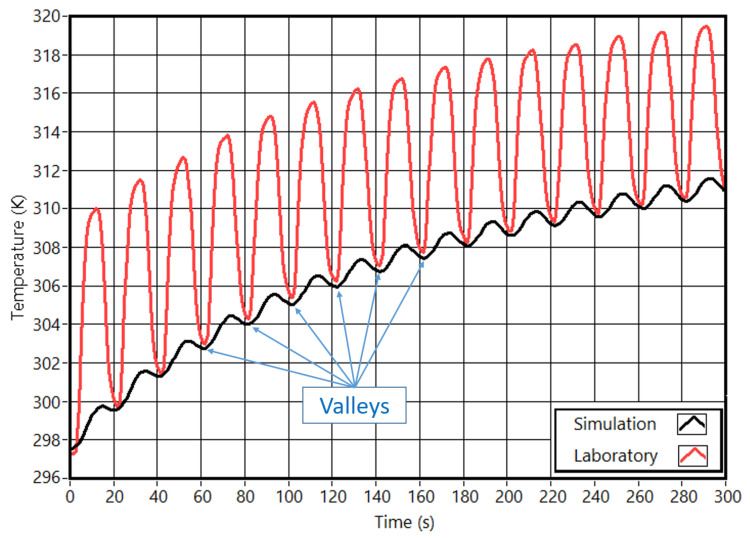
Obtained temperatures in laboratory and simulation at the center of slot 1.

**Figure 6 sensors-23-00325-f006:**
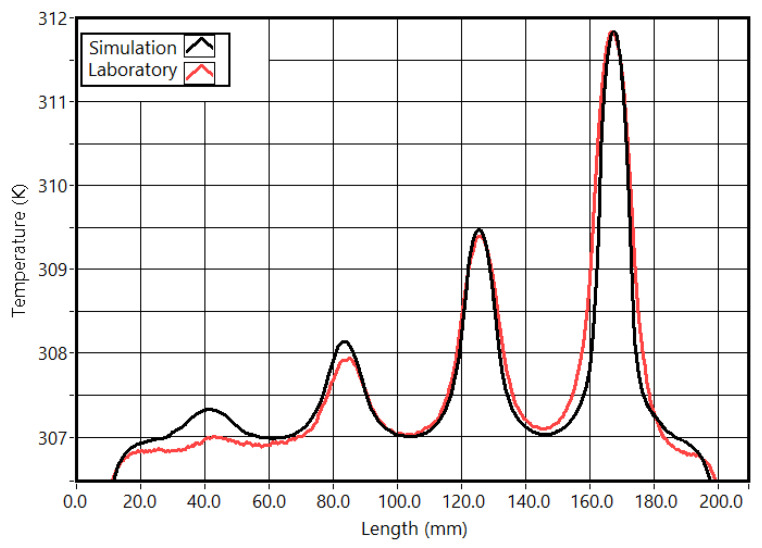
Temperature profiles at the end of the test in the stimulation surface (20 s).

**Figure 7 sensors-23-00325-f007:**
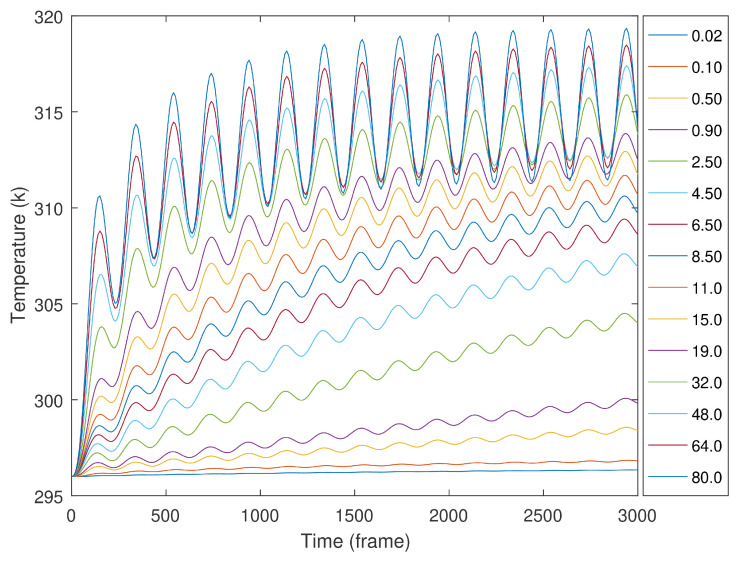
Temperature evolution at the center of slot 1 for cycle periods of 0.02, 0.10 up to 80 s.

**Figure 8 sensors-23-00325-f008:**
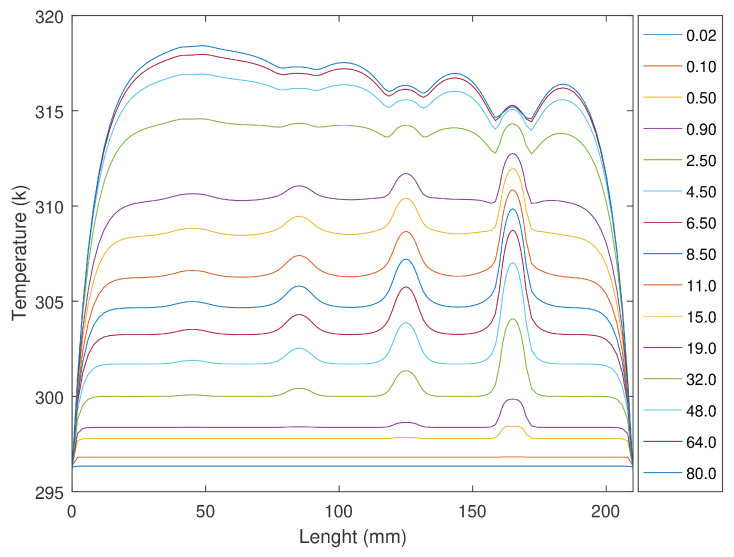
Temperature profiles in the stimulation surface after 15 cycles for cycle periods of 0.02, 0.10 up to 80 s.

**Figure 9 sensors-23-00325-f009:**
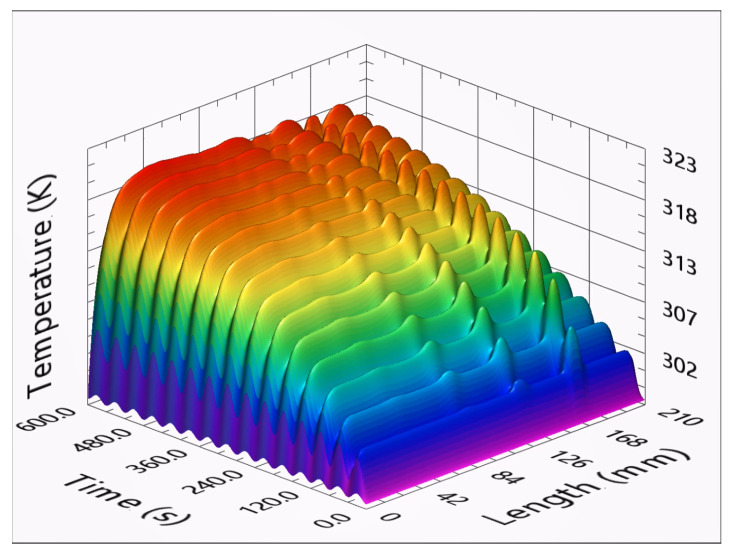
Temperature profiles throughout the entire test for a stimulation period of 40 s.

**Figure 10 sensors-23-00325-f010:**
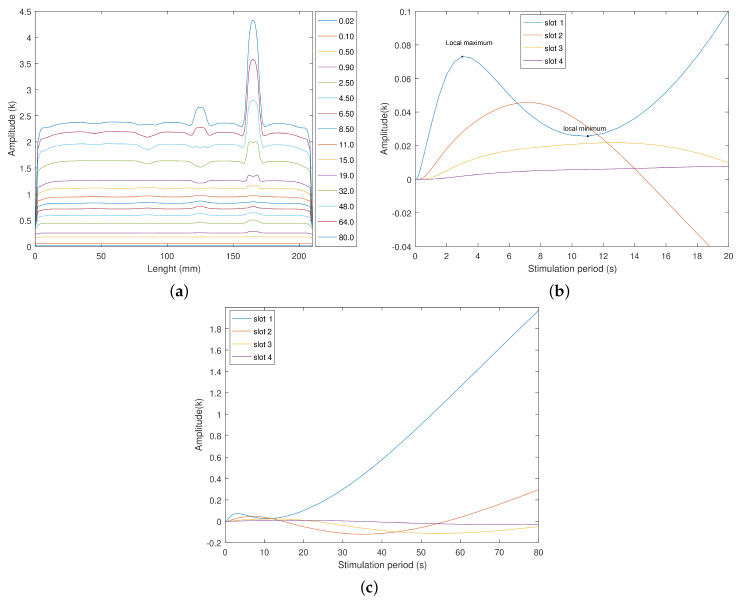
Amplitude response and Amplitude difference, function of the cycle period. (**a**) Amplitude responses from the stimulation surface after 15 cycles. (**b**) Amplitude difference between slots 1, 2, 3, and 4 and their sound areas, for cycle periods of 0.02 to 20 s. (**c**) Amplitude difference between the slot and its surroundings from 0.02 to 80 s per cycle.

**Figure 11 sensors-23-00325-f011:**
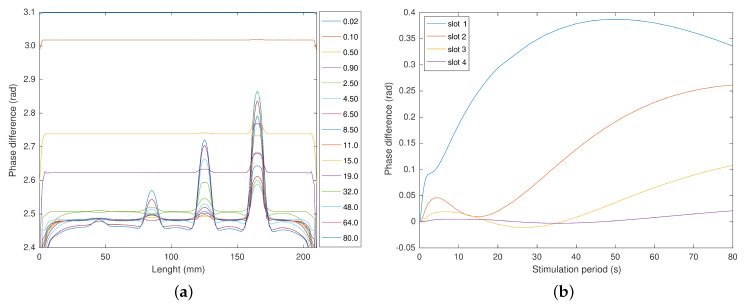
Phase delay and difference function of the cycle period. (**a**) Phase delay of the stimulation surface after 15 cycles. (**b**) Phase difference for periods from 0.02 to 80 s per cycle.

**Figure 12 sensors-23-00325-f012:**
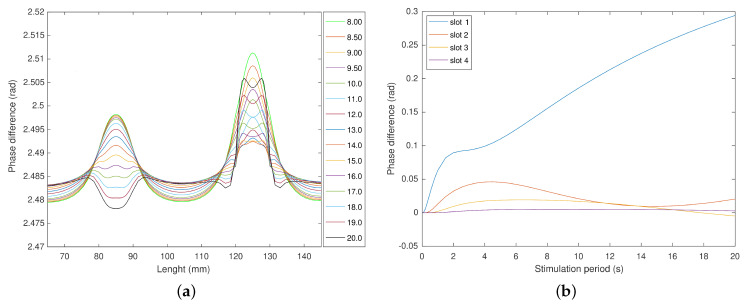
Detailed view of slots 2 and 3 for periods between 8 to 20 s. (**a**) Phase delay of slots 2 and 3 for stimulations ranging from 8 and 20 s. (**b**) Phase difference for periods from 0.02 to 20 s per cycle.

**Figure 13 sensors-23-00325-f013:**
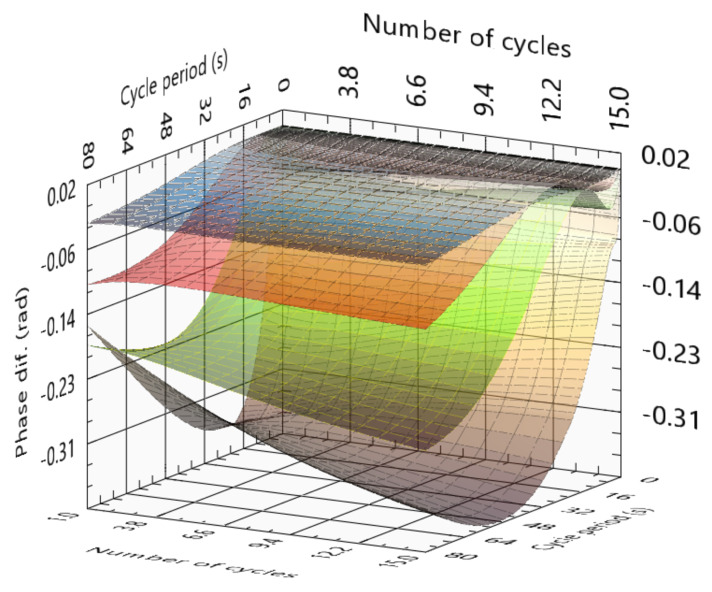
Phase difference function of cycle period, number of cycles, and for slots 1 to 4.

**Figure 14 sensors-23-00325-f014:**
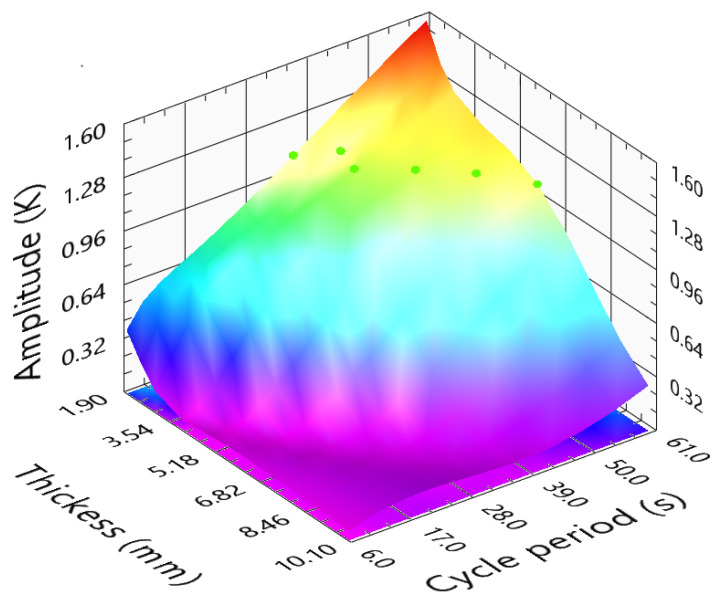
Amplitude difference function of PMMA sample thickness and cycle period.

**Figure 15 sensors-23-00325-f015:**
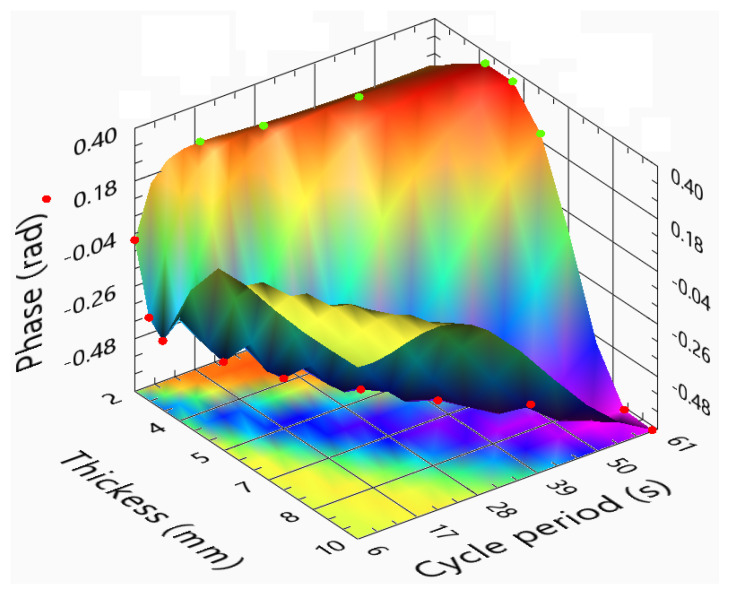
Phase difference function of the sample thickness and cycle period for PMMA samples.

**Figure 16 sensors-23-00325-f016:**
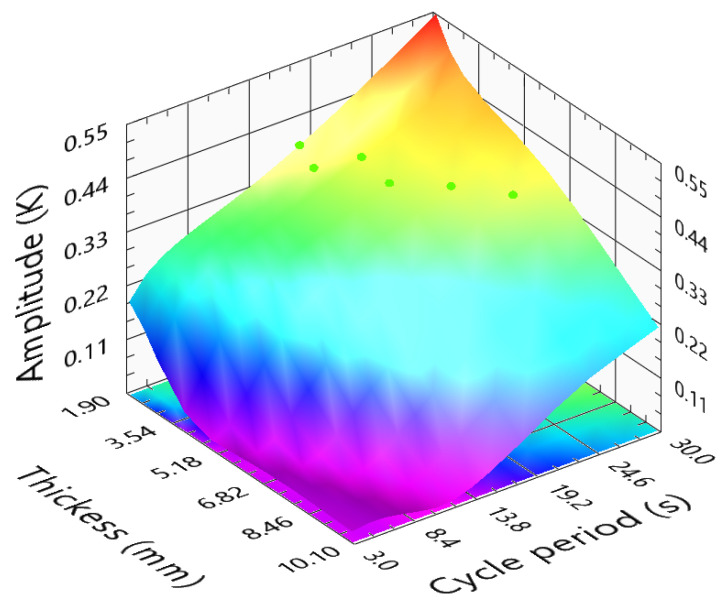
Amplitude difference function of the sample thickness and cycle period, for CFRP samples.

**Figure 17 sensors-23-00325-f017:**
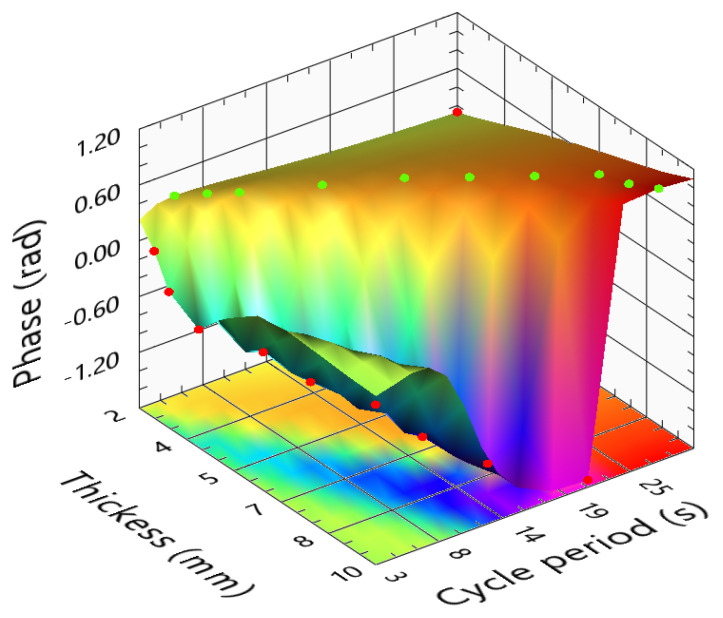
Phase difference function of the sample thickness and cycle period of CFRP samples.

**Figure 18 sensors-23-00325-f018:**
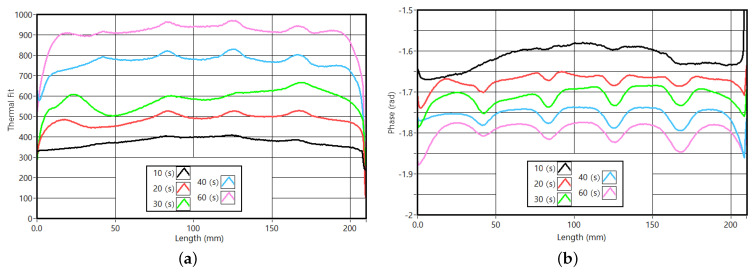
Results from the PMMA validation tests. (**a**) Average amplitude profiles. (**b**) Average phase profiles.

**Figure 19 sensors-23-00325-f019:**
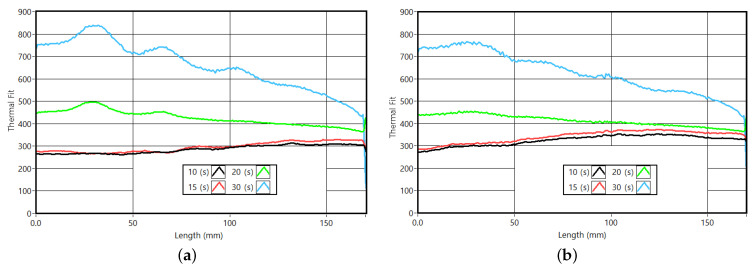
Vertical amplitude profiles for two slots. (**a**) Thermal profiles for the slot 2 mm thick. (**b**) Thermal profiles for the slots 4 mm thick.

**Figure 20 sensors-23-00325-f020:**
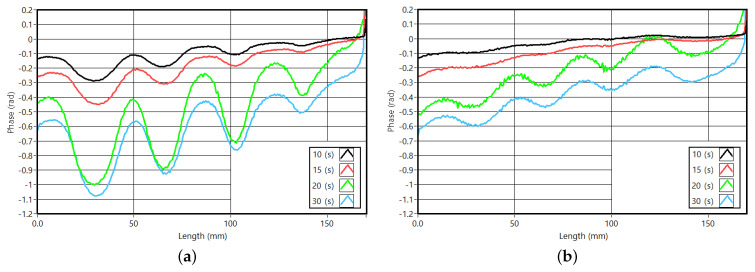
Vertical phase profiles for two rows of slots of the CFRP samples and various cycle periods. (**a**) Phase profiles for the slots 2 mm thick. (**b**) Phase profiles for the slots 4 mm thick.

**Table 1 sensors-23-00325-t001:** Reference temperatures from the center of the sound area and slot 1, obtained with various meshes.

Peak fig 5.4 and 5.5		
Mesh		
Num. Elements in *d*, *a/b*, and *c*	Sound Area	Slot 1
20 × 20 × 2	301.97281	306.77600
20 × 20 × 3	301.89808	306.73894
**20 × 20 × 4**	**301.92363**	**306.72390**
20 × 20 × 5	301.95672	306.72745
20 × 20 × 6	301.91589	306.71402
20 × 20 × 8	301.92080	306.71607
20 × 4 × 4	301.92380	307.29634
20 × 6 × 4	301.92373	306.94711
20 × 8 × 4	301.92369	306.84510
20 × 12 × 4	301.92366	306.76780
**20 × 16 × 4**	**301.92364**	**306.73865**
20 × 20 × 4	301.92363	306.72390
20 × 40 × 4	301.92362	306.70019
12 × 16 × 4	301.92388	306.69013
16 × 16 × 4	301.92372	306.72122
** 20 × 16 × 4 **	**301.92364**	**306.73865**
30 × 16 × 4	301.92356	306.75944
40 × 16 × 4	301.92353	306.76843

## Data Availability

Not applicable.

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
