# Peer review of "Lock-In Thermal Test Simulation, Influence, and Optimum Cycle Period for Infrared Thermal Testing in Non-Destructive Testing"

_sensors, 2022, doi:10.3390/s23010325_

Round 1

Reviewer 1 Report

The questions, issues and recommendations are as follows:

1.       In the abstract, line 8: The authors referred to the Figures 14-17, which is not proper. Be modified this section.

2.        The important results be added into the abstract section.

3.       The first paragraph in introduction section is too summary. Be added more explanations.

4.       Until fourth paragraph of introduction section, no referring has been done into the manuscript. Be added the related references.

5.       The mentioned novelty in lines 93-95 is too summary and unclear. More explanations be added.

6.       The conclusion section is too long. Be summarized that and only mentioned the necessary results.

7.       The number of used references is low. The new references, especially the papers, which have been published in 2020-2022, be added.

Author Response

Dear reviewer,

Thank you for the time dedicated to our work and the opportunity to improve it.
In the attached document are some notes and our reply to your comments. These are highlighted in green color.

Reviewer 2 Report

My remarks:

1. The authors didn't include the strong research implication.

2. The purpose of paper is not clear.

3. The authors need to provide their justification with regards to their novelty in the paper/ contribution to knowledge and cite more recent (2019-2022).

Author Response

(The authors gave the same response as above.)

Reviewer 3 Report

Dear authors,

The data provided for the article entitled " Lock-in Thermal Test Simulation, Influence, and optimum cycle period for infrared thermal testing in Non-destructive testing" (sensors-2074458) are interesting; however, I am offering some comments throughout the manuscript.

(1) Abstract: Readers would like to grab all of your key findings after having a look at the abstract. But the recent abstract is not conveying any experimental idea of the work at all; it looks like a combination of some conclusive statements. But the abstract must have key experimental findings.

(2) Line 7: All short forms are not abbreviated. It is recommended to use abbreviations first and then continue in a short form.

(3) The introduction part is unprofessional; there is a lack of consistency between lines and paragraphs; it really needs to be revised very carefully.

(4) The main research gap of the work is totally absent in the Introduction. The last paragraph of the Introduction should provide information (only) about the science gap in the previous studies and what motivates you to do this review with the objective of the study.

(5) Lines 12-54: Authors are recommended to use appropriate reference.

(6) Table 1: The authors are recommended to mention the unit of mesh.

(7) The entire conclusion must be re-written with conclusive findings and by retaining coherence.

(8) References should be in accordance with the journal template.

(9) I am not an English speaker, but I found many typos and grammatical errors throughout the manuscript. These must be corrected and revised.

Author Response

(The authors gave the same response as above.)

Round 2

Reviewer 2 Report

it is all OK